# Epigenetic clock analysis and increased plasminogen activator inhibitor-1 in high-functioning autism spectrum disorder

**Satoshi Okazaki[1], Ryo Kimura[2], Ikuo Otsuka[1], Yasuko Funabiki[3,4], Toshiya Murai[4], Akitoyo Hishimoto[1,5]***

**1** Department of Psychiatry, Kobe University Graduate School of Medicine, Kobe, Japan, **2** Department of Anatomy and Developmental Biology, Graduate School of Medicine, Kyoto University, Kyoto, Japan, **3** Department of Cognitive and Behavioral Science, Graduate School of Human and Environmental Studies, Kyoto University, Kyoto, Japan, **4** Department of Psychiatry, Graduate School of Medicine, Kyoto University, Kyoto, Japan, **5** Department of Psychiatry, Yokohama City University Graduate School of Medicine, Yokohama, Japan

* hishipon@yokohama-cu.ac.jp

**Data Availability Statement:** The DNA methylation dataset was deposited into the NCBI Gene

## Abstract

### Background

Autism spectrum disorder (ASD) is characterized by impaired social communication and behavioral problems. An increased risk of premature mortality has been observed in individuals with ASD. Therefore, we hypothesized that biological aging is accelerated in individuals with ASD. Recently, several studies have established genome-wide DNA methylation (DNAm) profiles as 'epigenetic clocks' that can estimate biological aging. In addition, ASD has been associated with differential DNAm patterns.

### Methods

We used two independent datasets from blood samples consisting of adult patients with high-functioning ASD and controls: the 1st cohort (38 ASD cases and 31 controls) and the 2nd cohort (6 ASD cases and 10 controls). We explored well-studied epigenetic clocks such as HorvathAge, HannumAge, SkinBloodAge, PhenoAge, GrimAge, and DNAm-based telomere length (DNAmTL). In addition, we investigated seven DNAm-based age-related plasma proteins, including plasminogen activator inhibitor-1 (PAI-1), and smoking status, which are the components of GrimAge.

### Results

Compared to controls, individuals with ASD in the 1st cohort, but not in the 2nd cohort, exhibited a trend for increased GrimAge acceleration and a significant increase of PAI-1 levels. A meta-analysis showed significantly increased PAI-1 levels in individuals with ASD compared to controls.

Expression Omnibus (GEO, https://www.ncbi.nlm.nih.gov/gds) as GSE109905.

**Funding:** This research was partly supported by grants from JSPS (Japan Society for the Promotion of Science) KAKENHI grant numbers 18K15483 and 21K07520 (S.O.) as well as 17H04249 and 21H02852 (A.H.). https://www.jsps.go.jp/j-grantsinaid/ The funder plays no role in the study design, data collection and analysis, decision to publish, or preparation of the manuscript.

**Competing interests:** The authors have declared that no competing interests exist.

## Conclusion

Our findings suggest there is no epigenetic age acceleration in the blood of individuals with ASD. However, this study provides novel evidence regarding increased plasma PAI-1 levels in individuals with high-functioning ASD. These findings suggest PAI-1 may be a biomarker for high-functioning ASD, however, larger studies based on epigenetic clocks and PAI-1 will be necessary to confirm these findings.

## Introduction

Autism spectrum disorder (ASD) is a heterogeneous neurodevelopmental disorder characterized by impaired social communication and behavioral problems that affects about 1% of the worldwide population [1]. Brain structural and diffusion magnetic resonance imaging studies have consistently found disrupted neuronal connectivity due to abnormal neuronal migration in individuals with ASD [2]. However, there is no useful biomarker to confirm the diagnosis of ASD or evaluate the efficacy of treatments [3]. In addition, an increased risk of premature mortality has been observed in individuals with ASD, which results from both natural causes and suicide [4–6]. Even after adjusting for comorbid psychiatric disorders such as intellectual disability, the mortality risk in individuals with ASD is elevated for natural causes but not for suicide [5]. Therefore, as a putative explanation for these observations, we propose a hypothesis that ASD is associated with accelerated biological aging.

Aging research has progressed markedly in the past decade. Among six potential estimators (i.e., epigenetic clocks; telomere length; transcriptomic-, proteomic-, and metabolomic-based predictors; composite markers), 'epigenetic clocks' are the most promising predictors for biological aging [7]. A number of epigenetic clocks have been established based on genome-wide DNA methylation (DNAm) profiles [8–13]. Such measures have been revealed an association of epigenetic age acceleration with various conditions including Down syndrome [14], Werner syndrome [15], Alzheimer's disease [16], schizophrenia [17, 18], bipolar disorder [19, 20], depressive disorder [21, 22], suicide [23], alcohol related disorders [24, 25], and posttraumatic stress disorder [26, 27].

Several epigenetic clocks have been established based on the specific CpG sites selected from different DNAm datasets. The first generation of epigenetic clocks such as DNAm HorvathAge [9], HannumAge [8], and SkinBloodAge [10] were developed to predict chronological age, and showed weak associations with clinical phenotypes [28]. Subsequently, PhenoAge [11] and GrimAge [12] were developed to capture physiological status and mortality, respectively. GrimAge is constructed from 10 clinical characteristics related to 'grim' news: chronological age, sex, seven DNAm-based age-related plasma proteins including plasminogen activator inhibitor-1 (PAI-1), and DNAm-based smoking status [12]. GrimAge components, such as DNAmPAI-1, can also be calculated [12]. GrimAge outperforms other epigenetic clocks in the prediction of age-related clinical phenotypes and all-cause mortality [29]. In contrast, DNAm telomere length (DNAmTL) was established to predict telomere length, which is also a biomarker of aging, from a DNAm profile [30].

Previous studies have reported altered DNAm profiles in the blood of individuals with ASD [31, 32]. Furthermore, Pediatric-Buccal-Epigenetic clock acceleration was found in buccal epithelium cells of children with ASD compared to typically developing children [13]. However epigenetic clock analysis using blood samples has not been reported in individuals with ASD.

Here, we performed epigenetic clock analysis using blood samples to compare individuals with ASD and controls. We investigated five measures of epigenetic age (HorvathAge, HannumAge, SkinBloodAge, PhenoAge, and GrimAge) and DNAmTL, as well as GrimAge components including DNAm-based age-related plasma proteins and smoking status. We focused on adults with high-functioning ASD, because the high mortality of ASD is associated with comorbid psychiatric disorders including intellectual disability [5].

## Materials and methods

### Participants

This study was conducted in accordance with the Declaration of Helsinki, and was approved by the ethics committee of Kyoto University Graduate School and Faculty of Medicine. Written informed consent was obtained from all participants.

The participants are from our previously published study [32]. Briefly, we used two independent cohorts consisting of adult patients with high-functioning ASD and controls; the 1st cohort comprised 38 ASD cases and 31 controls, and the 2nd cohort was six ASD cases and 10 controls. All participants were of Japanese descent and there are no racial differences in the samples. We recruited patients with ASD from the Department of Psychiatry at Kyoto University Hospital as well as healthy volunteers. The diagnosis of ASD was evaluated using the Diagnostic and Statistical Manual of Mental Disorders 5th Edition (DSM-5), followed by the Autism Diagnostic Observation Schedule (ADOS) instrument and Japanese version of the high-functioning Autism Spectrum Screening Questionnaire (ASSQ-R) [33, 34]. Intelligence quotient (IQ) was assessed according to the Wechsler Adult Intelligence Scale 3rd Edition (WAIS-III). We excluded participants with an IQ < 80, additional psychiatric/neurologic disorders, other medical disorders, history of cigarette smoking, or use of psychotropic medication for at least three months before the blood collection in both ASD and control groups. Age, sex, and IQ were matched between the ASD and control groups in both the 1st and 2nd cohorts. The 2nd cohort consisted of only male individuals.

### Epigenetic clocks

Genome-wide DNAm profiles of blood samples were measured using the Illumina Infinium 450 K platform as described elsewhere [32]. The Beta Mixture Quantile Dilation (BMIQ) method was used for array normalization [35]. We investigated HorvathAge, HannumAge, SkinBloodAge, PhenoAge, GrimAge, and DNAmTL, as well as GrimAge components, including DNAm-based plasma proteins and smoking status, using an online calculator (https://horvath.genetics.ucla.edu/html/dnamage/) [9, 12]. Epigenetic age acceleration was defined as the residual from regressing measurement values on chronological age. A positive or negative acceleration value indicates a higher or lower than expected clock value based on chronological age, respectively.

### Statistics

Statistical analyses were performed using R version 3.6.1 (The R Foundation for Statistical Computing, Vienna, Austria) with EZR version 1.41 (Jichi Medical University, Saitama, Japan) [36]. Differences in continuous variables between groups were analyzed using the Student's $t$-test or Man-Whitney U test, as appropriate. Multiple linear regression analysis was applied to address confounding factors such as age, sex, and phenotype. Meta-analysis was performed using a fixed effect model after confirming low heterogeneity between the 1st and 2nd cohorts by Cochran's Q test. The relationship between continuous variables was analyzed with

Spearman's rank correlation coefficient. Dummy variables were used as necessary. Statistical significance was defined as a two-tailed $p$-value < 0.05.

## Results

In the 1st cohort, there was no significant difference in epigenetic age acceleration between the ASD and control groups, although there was a trend for GrimAge acceleration ($p = 0.101$) (Table 1 and Fig 1). Next, we investigated GrimAge components including DNAm-based plasma proteins and smoking status and found significantly increased DNAmPAI-1 levels in individuals with ASD compared to controls ($p = 0.00385$) (Table 1 and Fig 2). Using multiple linear regression analysis to adjust confounding factors such as age and sex, there was still a significant difference in DNAmPAI-1 levels ($p = 0.0100$) (S1 Table).

In the 2nd cohort, there was no significant difference in epigenetic age acceleration, DNAm-based plasma proteins, and smoking status in individuals with ASD and the control group (Table 1, S1 and S2 Figs, as well as S1 Table).

**Table 1. Demographic and clinical characteristics as well as epigenetic clock acceleration.**

|  | 1st cohort | | | 2nd cohort | | |
|---|---|---|---|---|---|---|
|  | Control ($n = 31$) | ASD ($n = 38$) | $P$-value | Control ($n = 10$) | ASD ($n = 6$) | $P$-value |
| **Demographic and clinical characteristics** | | | | | | |
| Sex, male / female | 16 / 15 | 23 / 15 | 0.476[a] | 10 / 0 | 6 / 0 | NA |
| Age, mean (SD) | 26.7 (6.7) | 28.6 (6.5) | 0.244[b] | 31.2 (10.0) | 27.0 (7.0) | 0.383[b] |
| IQ, mean (SD) | 112.8 (12.1) | 107.8 (14.7) | 0.139[b] | 121.1 (10.87) | 110.0 (16.1) | 0.120[b] |
| ADOS total, median (IQR) | 1.00 (1.00, 2.00) | 6.00 (4.00, 8.50) | < 0.001[c] | 2.00 (0.25, 2.00) | 8.50 (7.25,12.75) | 0.00173[c] |
| ASSQR total, median (IQR) | 6.00 (2.50, 9.00) | 28.00 (22.00, 32.75) | < 0.001[c] | - | - | - |
| **Epigenetic age acceleration** | | | | | | |
| HorvathAge acceleration, mean (SD) | 0.33 (4.89) | −0.27 (4.31) | 0.591[b] | 1.42 (4.61) | −2.36 (8.60 | 0.267[b] |
| HannumAge acceleration, mean (SD) | −0.49 (3.88) | 0.40 (3.72) | 0.331[b] | 1.42 (2.69) | −2.37 (6.08) | 0.104[b] |
| SkinBloodAge acceleration, mean (SD) | −0.41 (3.10) | 0.34 (3.06) | 0.317[b] | 0.04 (2.27) | −0.07 (7.28) | 0.964[b] |
| PhenoAge acceleration, mean (SD) | −0.65 (5.11) | 0.53 (4.98) | 0.340[b] | 2.00 (4.77) | −3.34 (9.54) | 0.154[b] |
| GrimAge acceleration, mean (SD) | −0.46 (1.98) | 0.38 (2.16) | 0.101[b] | 0.31 (2.32) | −0.52 (2.65) | 0.518[b] |
| DNAmTL acceleration (kbp), mean (SD) | −0.01 (0.17) | 0.01 (0.16) | 0.600[b] | −0.03 (0.18) | 0.05 (0.09) | 0.306[b] |
| **GrimAge components** | | | | | | |
| DNAmADM (ng/mL), mean (SD) | 0.48 (0.02) | 0.48 (0.02) | 0.521[b] | 0.40 (0.03) | 0.38 (0.01) | 0.176[b] |
| DNAmB2M (ng/mL), mean (SD) | 2262.27 (89.03) | 2299.12 (106.30) | 0.129[b] | 2727.06 (149.12) | 2633.02 (110.27) | 0.203[b] |
| DNAmCystatinC (ng/mL), mean (SD) | 728.06 (22.38) | 736.34 (28.19) | 0.188[b] | 543.77 (39.15) | 535.37 (42.76) | 0.694[b] |
| DNAmGDF-15 (ng/mL), mean (SD) | 0.71 (0.11) | 0.73 (0.10) | 0.505[b] | 0.37 (0.14) | 0.29 (0.14) | 0.283[b] |
| DNAmLeptin (ng/mL), mean (SD) | −9.60 (2.13) | −9.32 (2.20) | 0.593[b] | 25.33 (2.39) | 23.35 (2.85) | 0.158[b] |
| DNAmPAI-1 (ng/mL), mean (SD) | 30.52 (1.55) | 31.66 (1.59) | **0.00385**[b] | 23.04 (3.17) | 23.07 (1.02) | 0.982[b] |
| DNAmTIMP-1 (ng/mL), mean (SD) | 28.88 (1.20) | 29.17 (1.08) | 0.280[b] | 26.86 (1.39) | 26.03 (1.27) | 0.253[b] |
| DNAmPACKYRS, mean (SD) | 12.78 (4.63) | 14.89 (5.67) | 0.101[b] | 8.33 (6.50) | 6.07 (5.39) | 0.488[b] |

ADM, adrenomedullin; ADOS, Autism Diagnostic Observation Schedule; ASD, autism spectrum disorder; ASSQ-R, Autism Spectrum Screening Questionnaire; B2M, beta-2-microglobulin; DNAm, DNA methylation; IQ, intelligence quotient; IQR, interquartile range; GDF-15, growth differentiation factor 15; PACKYRS, smoking pack-years; PAI-1, plasminogen activator inhibitor-1; SD, standard deviation; TIMP-1, tissue inhibitor of metalloproteinases-1.

Boldface type indicates statistical significance.

[a] $P$-value was calculated using the Fisher's exact test.

[b] $P$-value was calculated using the Student's $t$-test.

[c] $P$-value was calculated using the Mann-Whitney U test.

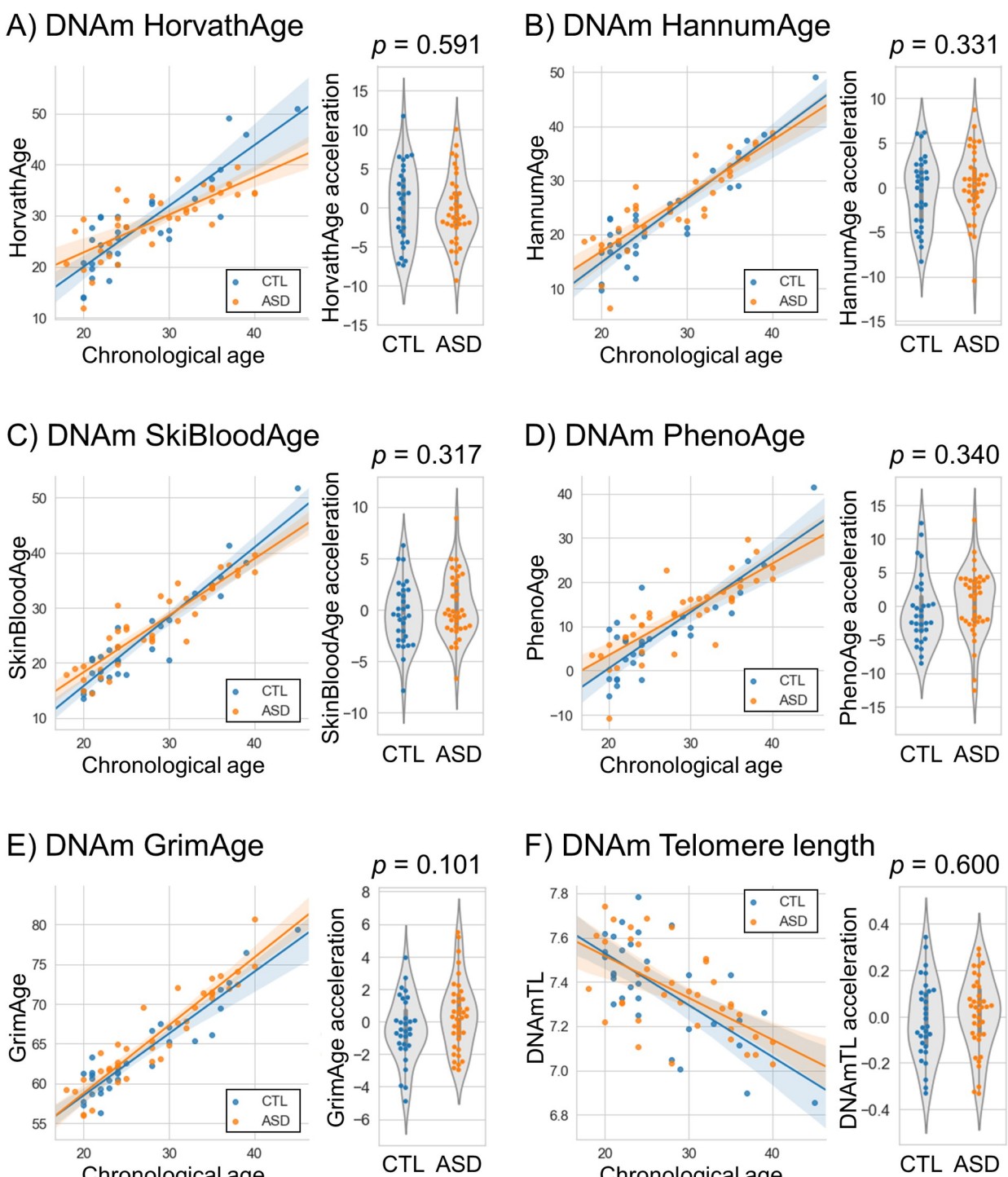

**Fig 1. Epigenetic clock analysis in the 1st cohort.** (**A**) HorvathAge, (**B**) HannmuAge, (**C**) SkinBloodAge, (**D**) PhenoAge, (**E**) GrimAge, and (**F**) DNAmTL. Scatter plots show epigenetic age vs. chronological age. Violin plots with dots show epigenetic age acceleration in the ASD and control groups. Student's *t*-tests were performed for comparisons between the groups. ASD, autism spectrum disorder; CTL, control; DNAmTL, DNA methylation-based telomere length.

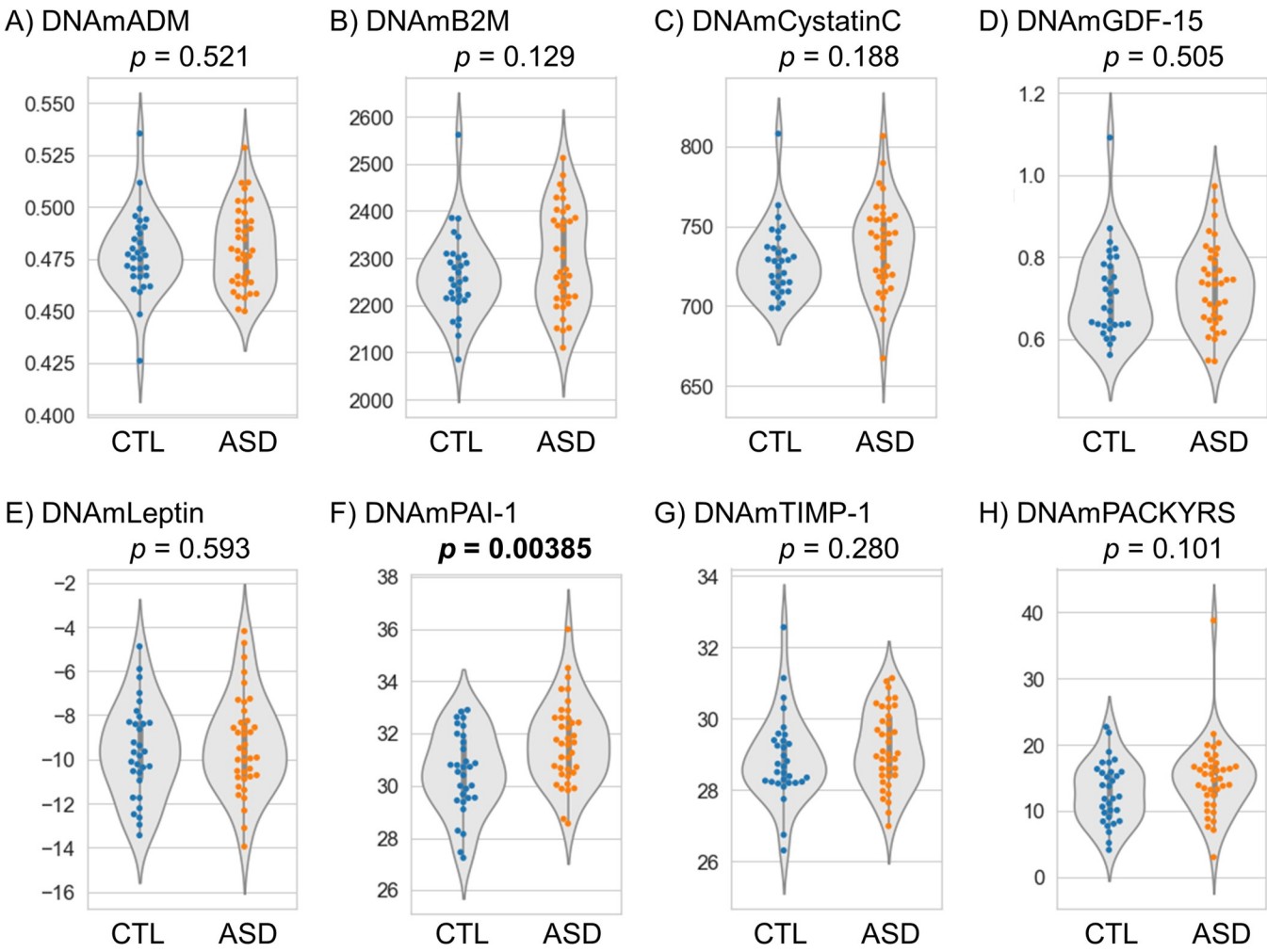

**Fig 2. DNA methylation-based age-related plasma proteins and smoking status in the 1st cohort.** (**A**) ADM, (**B**) B2M, (**C**) Cystatin C, (**D**) GDF-15, (**E**) Leptin, (**F**) PAI-1, (**G**) TIMP-1, and (**H**) PACKYRS. Student's t-tests were performed for comparisons between the groups. ADM, adrenomedullin; ASD, autism spectrum disorder; B2M, beta-2-microglobulin; CTL, control; GDF-15, growth differentiation factor 15; PACKYRS, smoking pack-years; PAI-1, plasminogen activator inhibitor-1; TIMP-1, tissue inhibitor of metalloproteinases-1.

In addition, we performed a meta-analysis of DNAmPAI-1. DNAmPAI-1 was significantly increased in individuals with ASD compared to controls ($p = 0.0044$) (S3 Fig), and showed the Bonferroni correction level (corrected significance was defined as $p$-value $< 0.05/7 = 0.0071$). The mean difference between the ASD and control groups indicated that individuals with ASD are predicted to have plasma PAI-1 levels 1.02 ng/mL higher than controls.

We additionally performed correlation analysis between epigenetic age accelerations/PAI-1 levels and IQ, ADOS, and ASSQ-R scores in individuals with ASD. We found that GrimAge acceleration was significantly positively correlated with ADOS total in both the 1st cohort ($rho = 0.415$, $p = 0.0131$) and the 2nd cohort ($rho = 0.886$, $p = 0.0333$), but did not show the Bonferroni correction level (corrected significance was defined as $p$-value $< 0.05/5 = 0.01$) in either cohort (S4 Fig).

## Discussion

To our knowledge, this is the first study to performed an epigenetic clock analysis of blood between individuals with high-functioning ASD and controls. We observed a non-significant

trend in GrimAge acceleration, and a significant difference in DNAmPAI-1 levels between ASD and control groups in the 1st, but not the 2nd, cohort. A meta-analysis showed significantly increased DNAmPAI-1 levels in individuals with ASD compared to controls. No biomarker is currently available for confirming the diagnosis and efficacy of treatments for ASD [3]. Our findings suggest that plasma PAI-1 levels could represent potential blood biomarkers for ASD.

We additionally observed a modest significant correlation between GrimAge acceleration and ADOS total in both the 1st and 2nd cohorts; however, these results did not show Bonferroni correction level. Therefore, it was not possible to draw a definitive conclusion. Further studies with a larger sample size are required.

McEwen LM, et al previously reported that epigenetic age is accelerated in children with ASD [13], which is inconsistent with our present findings. They used the Pediatric-Buccal-Epigenetic clock based on buccal epithelium cells. In contrast, we used other epigenetic clocks based on blood samples. The differences between our and their findings may be due to difference in the type of tissues used for the development of each epigenetic clock.

PAI-1 is a major physiological inhibitor of tissue plasminogen activator (tPA) in plasma and is increased in various clinical situations related to ischemic cardiovascular events and senescence [37, 38]. Plasma PAI-1 levels are increased by hyperglycemia and hyperinsulinemia [39]. Recent epidemiological studies have shown that maternal diabetes is associated with an increased prevalence of ASD [40, 41]. Based on this finding, Hoirisch-Clapauch et al. hypothesized that neuronal dis-connectivity due to abnormal neuronal migration underlies the development of ASD, and that fetal hyperglycemia increases PAI-1 levels causing the inhibition of tPA activity for reelin glycoprotein, which guides neurons and glial cells from the ventricular zone to the cortex [42]. Our finding of increased DNAmPAI-1 in individuals with ASD may support this hypothesis. Moreover, plasma PAI-1 levels were significantly higher in individuals with ASD and both regression and a developmental delay compared to individuals with ASD without regression, or individuals with ASD and regression without a developmental delay [43]. In contrast, no association was observed between ASD and the 4G/5G sequence polymorphism in the PAI-1 gene promoter that influences the level of plasma PAI-1 [44]. Further research is required to elucidate the role of PAI-1 in the pathophysiology of ASD.

There are several limitations to the present study. First, the sample size of the cohorts analyzed was relatively small for a definitive conclusion. Independent studies with larger sample sizes are required in future work. Second, DNAmPAI-1 studied here is a prediction of plasma PAI-1 levels based on the DNAm profile, and it is necessary to confirm the actual plasma PAI-1 protein concentration. Third, this study included only Japanese individuals from a single university hospital. Fourth, information about other diseases or metabolic problems such as body mass index was lacking. Finally, only blood samples were analyzed. Future studies using other tissue and cell types, such as neuronal/glial cells and buccal epithelium cells, are required.

## Conclusions

We explored the DNAm-based epigenetic age and age-related plasma proteins, and identified increased DNAmPAI-1 in individuals with high-functioning ASD. Our findings suggest there is no epigenetic age acceleration of blood in ASD. However, this study provides novel evidence regarding increased plasma PAI-1 levels in individuals with high-functioning ASD. These findings may provide applicable biomarkers for high-functioning ASD, although, larger studies based on epigenetic clocks and PAI-1 will be necessary to confirm these findings.

## Supporting information

**S1 Fig. Epigenetic clock analysis in the 2nd cohort.** (**A**) HorvathAge, (**B**) HannmuAge, (**C**) SkinBloodAge, (**D**) PhenoAge, (**E**) GrimAge, and (**F**) DNAmTL. Scatter plots show the epigenetic age vs. chronological age. Violin plots with dots show epigenetic age acceleration in the ASD and control groups. Student's *t*-tests were performed for comparisons between the groups. ASD, autism spectrum disorder; CTL, control; DNAmTL, DNA methylation-based telomere length.
(TIF)

**S2 Fig. DNA methylation-based age-related plasma proteins and smoking status in the 2nd cohort.** (**A**) ADM, (**B**) B2M, (**C**) Cystatin C, (**D**) GDF-15, (**E**) Leptin, (**F**) PAI-1, (**G**) TIMP-1, and (**H**) PACKYRS. Student's t-tests were performed for comparisons between the groups. ADM, adrenomedullin; ASD, autism spectrum disorder; B2M, beta-2-microglobulin; CTL, control; GDF-15, growth differentiation factor 15; PACKYRS, smoking pack-years; PAI-1, plasminogen activator inhibitor-1; TIMP-1, tissue inhibitor of metalloproteinases-1.
(TIF)

**S3 Fig. Meta-analyses of DNAmPAI-1.** Low heterogeneity was observed between the 1st and 2nd cohorts (Cochran's Q test *p* = 0.33). A significant difference was observed between the ASD and control groups using a fixed effect model (*p* = 0.0044). ASD, autism spectrum disorder; CI, confidence interval; CTL, control; PAI-1, plasminogen activator inhibitor-1; SD, standard deviation.
(TIF)

**S4 Fig. Relationship between epigenetic clock acceleration/PAI-1 and IQ, ADOS, and ASSQ-R scores in individuals with ASD.** Scatter plots show epigenetic clock acceleration/PAI-1 vs. medical condition. The relationship was analyzed with Spearman's rank correlation coefficient. ADOS, the Autism Diagnostic Observation Schedule; ASD, autism spectrum disorder; ASSQ-R, the high-functioning Autism Spectrum Screening Questionnaire; CTL, control; DNAmTL, DNA methylation-based telomere length; IQ, intelligence quotient; PAI-1, plasminogen activator inhibitor-1.
(TIF)

**S1 Table. Multiple-linear regression analysis of DNAmPAI-1.** B, unstandardized partial regression coefficient; DNAmPAI-1, DNA methylation-based plasminogen activator inhibitor-1; SE, standard error. Multiple linear regression analysis was performed with PAI-1 as the response variable and phenotype, sex, and age as the explanatory variables. Dummy variables were used as follows: phenotype, control = 0 and autism spectrum disorder = 1; sex, male = 0 and female = 1. Boldface type indicates statistical significance.
(DOCX)

## Author Contributions

**Conceptualization:** Akitoyo Hishimoto.

**Data curation:** Ryo Kimura.

**Formal analysis:** Satoshi Okazaki, Ikuo Otsuka.

**Investigation:** Ryo Kimura.

**Project administration:** Ryo Kimura.

**Resources:** Yasuko Funabiki, Toshiya Murai.

**Supervision:** Akitoyo Hishimoto.

**Writing – original draft:** Satoshi Okazaki.

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
