## [Decision Letter · Decision Letter 0]

20 Dec 2021

PONE-D-21-33732Epigenetic clock analysis in high-functioning autism spectrum disorderPLOS ONE

Dear Dr. Hishimoto,

Thank you for submitting your manuscript to PLOS ONE. After careful consideration, we feel that it has merit but does not fully meet PLOS ONE’s publication criteria as it currently stands. Therefore, we invite you to submit a revised version of the manuscript that addresses the points raised during the review process. Two reviewers assessed the paper, and gave important suggestions.The authors are requested to revise the manuscript considering all the points raised by the referees.Please submit your revised manuscript by Feb 03 2022 11:59PM. If you will need more time than this to complete your revisions, please reply to this message or contact the journal office at plosone@plos.org. Please include the following items when submitting your revised manuscript:A rebuttal letter that responds to each point raised by the academic editor and reviewer(s). You should upload this letter as a separate file labeled 'Response to Reviewers'.A marked-up copy of your manuscript that highlights changes made to the original version. You should upload this as a separate file labeled 'Revised Manuscript with Track Changes'.An unmarked version of your revised paper without tracked changes. You should upload this as a separate file labeled 'Manuscript'.

We look forward to receiving your revised manuscript.

Kind regards,

Tadafumi Kato

Academic Editor

PLOS ONE

Journal Requirements:

"This research was partly supported by grants from JSPS KAKENHI grant numbers 18K15483 and 21K07520 (S.O.) as well as 17H04249 and 21H02852 (A.H.)."

"This research was partly supported by grants from JSPS (Japan Society for the Promotion of Science) KAKENHI grant numbers 18K15483 and 21K07520 (S.O.) as well as 17H04249 and 21H02852 (A.H.). https://www.jsps.go.jp/j-grantsinaid/

The funder plays no role in the study design, data collection and analysis, decision to publish, or preparation of the manuscript."

Reviewers' comments:

Reviewer's Responses to Questions

**Comments to the Author**

1. Is the manuscript technically sound, and do the data support the conclusions?

Reviewer #1: No

Reviewer #2: Yes

2. Has the statistical analysis been performed appropriately and rigorously? 

Reviewer #1: No

Reviewer #2: Yes

3. Have the authors made all data underlying the findings in their manuscript fully available?

Reviewer #1: Yes

Reviewer #2: Yes

4. Is the manuscript presented in an intelligible fashion and written in standard English?

Reviewer #1: Yes

Reviewer #2: Yes

5. Review Comments to the Author

Reviewer #1: In this manuscript, the authors investigated epigenetic age acceleration associated with autism spectrum disorder (ASD) using two independent datasets from blood samples consisting of adult patients with high-functioning ASD and controls. As results, they could not find significant difference in epigenetic age acceleration between ASD and controls in both cohorts. On the other hand, they found a significant increase of DNA methylation-based plasminogen activator inhibitor-1 (DNAmPAI-1) level in individuals with ASD compared to controls in the 1st cohort but not in the 2nd cohort. Finally, they performed a meta-analysis of DNAmPAI-1, and confirmed the significant increase of DNAmPAI-1 in ASD. Furthermore, they discussed the association PAI-1 and ASD.

The content of this manuscript was not suitable for the title and objective.

If they focus the association PAI-1 and ASD, they should change the title and objective of this manuscript.

If they focus epigenetic age acceleration associated with ASD, they should perform meta-analysis of epigenetic clock analysis and discuss the differences between the previous report which found the epigenetic age acceleration of ASD (McEwen LM, et al., 2020) and their study.

In addition, they should consider other factors affected the alteration of DNA methylation such as medical condition (ie. metabolic problems), prescription, real smoking condition and so on.

Reviewer #2: The authors investigated whether biological aging is observed in adult high-functioning ASD individuals based on age prediction from genome-wide DNA methylation profiles using DNAs extracted from whole blood. The participants in the two independent cohorts (1st: 38 ASD cases vs. 31 controls; 2nd: 6 vs. 10) were Japanese, non-smokers, recruited from a single university hospital, had no comorbidity, no psychotropic medication in at least three months prior to blood collection, and were phenotyped appropriately on the standardized assessment tools. The controls were recruited from the same site and were matched for age, sex, and IQ. The methylation analysis were also performed using the well-documented method, and the statistics were processed appropriately. As a result, they observed a non-significant trend in GrimAge acceleration and a significant difference in DNAmPAI-1 levels between ASD and control groups in the 1st cohort but not in the 2nd cohort. The meta-analysis showed significantly increased DNAmPAI-1 levels in individuals with ASD compared to controls. Although this study has some limitations such as small sample size and lack of plasma PAI-1 levels, to the best of my knowledge, it is the first report of epigenetic clock analysis of whole blood DNA in ASD.

1, P3 L12 "HorvahAge" -> "HorvathAge"

2, P16 L5 The fact that there are no racial differences in the samples is also a strength, so it would be good to describe it in the method.

6. PLOS authors have the option to publish the peer review history of their article (what does this mean?). If published, this will include your full peer review and any attached files.

Reviewer #1: No

Reviewer #2: **Yes: **Takafumi Shimada

---

## [Author Response · Author response to Decision Letter 0]

7 Jan 2022

We greatly appreciate the review of our paper entitled “Epigenetic clock analysis and increased plasminogen activator inhibitor-1 in high-functioning autism spectrum disorder”: Manuscript ID PONE-D-21-33732_R1”. The comments from the reviewer were helpful to improve and polish our paper. We have revised our paper according to the comments and marked added texts made in the revised manuscript in red for better readability. Point by point replies are below for the reviewers.

Reviewer #1: 

In this manuscript, the authors investigated epigenetic age acceleration associated with autism spectrum disorder (ASD) using two independent datasets from blood samples consisting of adult patients with high-functioning ASD and controls. As results, they could not find significant difference in epigenetic age acceleration between ASD and controls in both cohorts. On the other hand, they found a significant increase of DNA methylation-based plasminogen activator inhibitor-1 (DNAmPAI-1) level in individuals with ASD compared to controls in the 1st cohort but not in the 2nd cohort. Finally, they performed a meta-analysis of DNAmPAI-1, and confirmed the significant increase of DNAmPAI-1 in ASD. Furthermore, they discussed the association PAI-1 and ASD.

The content of this manuscript was not suitable for the title and objective.

If they focus the association PAI-1 and ASD, they should change the title and objective of this manuscript.

Response: Thank you for your valuable comment. We revised the title as follows:

- Epigenetic clock analysis and increased plasminogen activator inhibitor-1 in high-functioning autism spectrum disorder

(Page 1)

If they focus epigenetic age acceleration associated with ASD, they should perform meta-analysis of epigenetic clock analysis and discuss the differences between the previous report which found the epigenetic age acceleration of ASD (McEwen LM, et al., 2020) and their study.

Response: Thank you for your important pointing. As you indicated, McEwen LM, et al previously reported that epigenetic age is accelerated in children with ASD. Our results appear to be inconsistent with theirs. They used the Pediatric-Buccal-Epigenetic clock based on buccal epithelium cells. In contrast, we used other epigenetic clocks based on blood samples. The differences between our and their findings may be due to difference in the type of tissues used for the development of each epigenetic clock. We revised the Discussion as follows:

- McEwen LM, et al previously reported that epigenetic age is accelerated in children with ASD [13], which is inconsistent with our present findings. They used the Pediatric-Buccal-Epigenetic clock based on buccal epithelium cells. In contrast, we used other epigenetic clocks based on blood samples. The differences between our and their findings may be due to difference in the type of tissues used for the development of each epigenetic clock.

(Page 14)

- Finally, only blood samples were analyzed. Future studies using other tissue and cell types, such as neuronal/glial cells and buccal epithelium cells, are required.

(Page 16)

In addition, they should consider other factors affected the alteration of DNA methylation such as medical condition (ie. metabolic problems), prescription, real smoking condition and so on.

Response: Thank you for your valuable comment. We additionally performed correlation analysis between epigenetic age accelerations/PAI-1 levels and IQ, ADOS, and ASSQ-R scores in individuals with ASD. We found that GrimAge acceleration was significantly positively correlated with ADOS total in both the 1st cohort (rho = 0.415, p = 0.0131) and the 2nd cohort (rho = 0.886, p = 0.0333), but did not show the Bonferroni correction level (corrected significance was defined as p-value < 0.05/5 = 0.01) in either cohort. We have revised the manuscript as follows:

- The relationship between continuous variables was analyzed with Spearman’s rank correlation coefficient.

(Page 9)

- We additionally performed correlation analysis between epigenetic age accelerations/PAI-1 levels and IQ, ADOS, and ASSQ-R scores in individuals with ASD. We found that GrimAge acceleration was significantly positively correlated with ADOS total in both the 1st cohort (rho = 0.415, p = 0.0131) and the 2nd cohort (rho = 0.886, p = 0.0333), but did not show the Bonferroni correction level (corrected significance was defined as p-value < 0.05/5 = 0.01) in either cohort (S4 Fig).

(Page 13)

- We additionally observed a modest significant correlation between GrimAge acceleration and ADOS total in both the 1st and 2nd cohorts; however, these results did not show Bonferroni correction level. Therefore, it was not possible to draw a definitive conclusion. Further studies with a larger sample size are required.

(Page 14) 

- S4 Fig. Relationship between epigenetic clock acceleration/PAI-1 and IQ, ADOS, and ASSQ-R scores in individuals with ASD.

Scatter plots show epigenetic clock acceleration/PAI-1 vs. IQ, ADOS, and ASSQ-R scores. The relationship was analyzed with Spearman’s rank correlation coefficient.

ADOS, the Autism Diagnostic Observation Schedule; ASD, autism spectrum disorder; ASSQ-R, the high-functioning Autism Spectrum Screening Questionnaire; CTL, control; DNAmTL, DNA methylation-based telomere length; IQ, intelligence quotient; PAI-1, plasminogen activator inhibitor-1.

(Page 21)

Response: Regarding other medical conditions, our samples have no real smoking history and no use psychotropic medication. We revised the Methods as follows. Unfortunately, we don't have any information about other diseases or metabolic problems including body mass index. We added the limitation in the Discussion as follows. 

- We excluded participants with an IQ < 80, additional psychiatric/neurologic disorders, other medical disorders, history of cigarette smoking, or use of psychotropic medication for at least three months before the blood collection in both ASD and control groups.

(Page 7)

- Fourth, information about other diseases or metabolic problems such as body mass index was lacking.

(Page 16)

Reviewer #2: 

The authors investigated whether biological aging is observed in adult high-functioning ASD individuals based on age prediction from genome-wide DNA methylation profiles using DNAs extracted from whole blood. The participants in the two independent cohorts (1st: 38 ASD cases vs. 31 controls; 2nd: 6 vs. 10) were Japanese, non-smokers, recruited from a single university hospital, had no comorbidity, no psychotropic medication in at least three months prior to blood collection, and were phenotyped appropriately on the standardized assessment tools. The controls were recruited from the same site and were matched for age, sex, and IQ. The methylation analysis were also performed using the well-documented method, and the statistics were processed appropriately. As a result, they observed a non-significant trend in GrimAge acceleration and a significant difference in DNAmPAI-1 levels between ASD and control groups in the 1st cohort but not in the 2nd cohort. The meta-analysis showed significantly increased DNAmPAI-1 levels in individuals with ASD compared to controls. Although this study has some limitations such as small sample size and lack of plasma PAI-1 levels, to the best of my knowledge, it is the first report of epigenetic clock analysis of whole blood DNA in ASD.

1, P3 L12 "HorvahAge" -> "HorvathAge"

Response: Thank you for your comment. We revised the Abstract as follows:

- ... HorvathAge, ...

(Page 2)

2, P16 L5 The fact that there are no racial differences in the samples is also a strength, so it would be good to describe it in the method.

Response: Thank you for your valuable comment. We revised the Methods as follows:

- All participants were of Japanese descent and there are no racial differences in the samples.

(Page 7)

Journal Requirements:

2. You indicated that you had ethical approval for your study. In your Methods section, please ensure you have also stated whether you obtained consent from parents or　guardians of the minors included in the study or whether the research ethics committee or IRB specifically waived the need for their consent.

Response: Our cohorts consist of adult patients with high-functioning ASD and controls. We revised the Methods as follows:

- Briefly, we used two independent cohorts consisting of adult patients with high-functioning ASD and controls; the 1st cohort comprised 38 ASD cases and 31 controls, and the 2nd cohort was six ASD cases and 10 controls.

(Page 7)

---

## [Decision Letter · Decision Letter 1]

20 Jan 2022

Epigenetic clock analysis and increased plasminogen activator inhibitor-1 in high-functioning autism spectrum disorder

PONE-D-21-33732R1

Dear Dr. Hishimoto,

We’re pleased to inform you that your manuscript has been judged scientifically suitable for publication and will be formally accepted for publication once it meets all outstanding technical requirements.

Kind regards,

Tadafumi Kato

Academic Editor

PLOS ONE

Additional Editor Comments (optional):

Reviewers' comments:

Reviewer's Responses to Questions

**Comments to the Author**

1. If the authors have adequately addressed your comments raised in a previous round of review and you feel that this manuscript is now acceptable for publication, you may indicate that here to bypass the “Comments to the Author” section, enter your conflict of interest statement in the “Confidential to Editor” section, and submit your "Accept" recommendation.

Reviewer #1: All comments have been addressed

Reviewer #2: All comments have been addressed

2. Is the manuscript technically sound, and do the data support the conclusions?

Reviewer #1: Yes

Reviewer #2: (No Response)

3. Has the statistical analysis been performed appropriately and rigorously? 

Reviewer #1: Yes

Reviewer #2: (No Response)

4. Have the authors made all data underlying the findings in their manuscript fully available?

Reviewer #1: Yes

Reviewer #2: (No Response)

5. Is the manuscript presented in an intelligible fashion and written in standard English?

Reviewer #1: Yes

Reviewer #2: (No Response)

6. Review Comments to the Author

Reviewer #1: (No Response)

Reviewer #2: (No Response)

7. PLOS authors have the option to publish the peer review history of their article (what does this mean?). If published, this will include your full peer review and any attached files.

Reviewer #1: No

Reviewer #2: **Yes: **Takafumi Shimada

---

## [Editor Report · Acceptance letter]

24 Jan 2022

PONE-D-21-33732R1 

Epigenetic clock analysis and increased plasminogen activator inhibitor-1 in high-functioning autism spectrum disorder 

Dear Dr. Hishimoto:

I'm pleased to inform you that your manuscript has been deemed suitable for publication in PLOS ONE. Congratulations! Your manuscript is now with our production department. 

Kind regards, 

on behalf of

Dr. Tadafumi Kato 

Academic Editor

PLOS ONE